# Epidemiology and Genomic Analysis of Equine Encephalosis Virus Detected in Horses with Clinical Signs in South Africa, 2010–2017

**DOI:** 10.3390/v13030398

**Published:** 2021-03-02

**Authors:** Jumari Snyman, Otto Koekemoer, Antoinette van Schalkwyk, Petrus Jansen van Vuren, Louwtjie Snyman, June Williams, Marietjie Venter

**Affiliations:** 1Centre for Viral Zoonoses, Department Medical Virology, University of Pretoria, Pretoria 0001, South Africa; jumari.steyn@gmail.com (J.S.); Petrus.Jansenvanvuren@csiro.au (P.J.v.V.); 2Agricultural Research Council, Onderstepoort Veterinary Research, Onderstepoort 0110, South Africa; otto@obpvaccines.co.za (O.K.); VanSchalkwykA1@arc.agric.za (A.v.S.); 3National Health Laboratory Service, Centre for Emerging Zoonotic and Parasitic Diseases, National Institute for Communicable Diseases, Johannesburg 2131, South Africa; 4Faculty of Veterinary Science， Department Veterinary Tropical Disease, University of Pretoria, Onderstepoort 0110, South Africa; lokisnyman@gmail.com; 5Department of Paraclinical Science, Section Pathology, Faculty of Veterinary Science, University of Pretoria, Onderstepoort 0110, South Africa; june.williams@up.ac.za

**Keywords:** EEV, next generation sequencing, febrile disease, equids, neurological signs, orbivirus

## Abstract

Equine encephalosis virus (EEV) is a neglected virus endemic to South Africa and is considered to generally result in mild disease in equines. Specimens were analyzed from live horses that presented with undefined neurological, febrile, or respiratory signs, or sudden and unexpected death. Between 2010 and 2017, 111 of 1523 (7.3%) horse samples tested positive for EEV using a nested real-time reverse transcriptase polymerase chain reaction (rRT-PCR). Clinical signs were reported in 106 (7.2%) EEV positive and 1360 negative horses and included pyrexia (77/106, 72.6%), icterus (20/106, 18.9%) and dyspnea (12/106, 11.3%). Neurological signs were inversely associated with EEV infection (OR < 1, *p* < 0.05) relative to EEV negative cases despite a high percentage of animals presenting with neurological abnormalities (51/106, 48.1%). Seventeen of the EEV positive horses also had coinfections with either West Nile (5/106, 4.7%), Middelburg (4/106, 3.8%) or African Horse sickness virus (8/106, 7.6%). To investigate a possible genetic link between EEV strains causing the observed clinical signs in horses, the full genomes of six isolates were compared to the reference strains. Based on the outer capsid protein (VP2), serotype 1 and 4 were identified as the predominant serotypes with widespread reassortment between the seven different serotypes.

## 1. Introduction

Equine encephalosis virus (EEV) belongs to the genus *Orbivirus* in the family *Reoviridae* and is endemic to South Africa. It is the cause of equine encephalosis that is often described as a mild form of African horse sickness (AHS), due to the mild febrile and non-fatal clinical signs [1]. The first EEV isolation was performed in 1967 and the virus was thought to be restricted to southern Africa, yet it was recently detected in Israel [1,2,3] and India [4]. Despite a significantly high seroprevalence for the virus in equids in southern Africa, detailed studies on the disease and virus are lacking [5]. Equine encephalosis virus is closely related to other orbiviruses such as BTV and African horse sickness virus (AHSV) [6], with the latter responsible for devastating outbreaks of severe cardiac, mixed, and pulmonary forms of disease in horses [7]. Bluetongue virus (BTV) represents the best characterized of the orbiviruses, both genetically and structurally, due to the great economic burden associated with bluetongue disease in livestock [8]. Both AHSV and BTV are endemic in South Africa, causing annual outbreaks locally as well as numerous epizootics in the Mediterranean region [9], Europe [10], South-West Asia [11], and most recently in Thailand [12]. This raises the concern of emergence of these orbiviruses in areas where the *Culicoides* vector occurs. Similar to other orbiviruses, EEV is a non-infectious arbovirus transmitted by hematophagous biting midges in the genus *Culicoides* (Diptera: Ceratopogonidae) [13,14,15,16]. The predominant EEV-associated *Culicoides* species—*Culicoides imicola* and *Culicoides bolitinos*—are also important vectors for other economically important arboviruses such as BTV and AHSV. This allows for co-circulation of the viruses and possible coinfections. Genome packaging of the individual species is specific and not known to reassort between species, although reassortment is described within the species [17,18].

Orbiviruses are 10-segmented double-stranded non-enveloped RNA viruses encoding 7 structural (VP1 to VP7) and 4 non-structural (NS1 to NS4) proteins [7,19,20]. The seven structural proteins are organized into an outer layer and inner shell [7,21,22]. The outer layer consists of VP2 (segment 2) and VP5 (segment 6) that are responsible for virus-receptor binding to target mammalian cells and viral fusion facilitation respectively [22]. Both VP2 and VP5 are highly variable and used to determine virus serotype [22,23]. The inner core layer consists of VP3 (segment3) and VP7 (segment 7), serogroup-specific antigens, which in turn are responsible for insect cell infection [22,23,24]. Viral transcriptase is facilitated by VP1 (segment 1), VP4 (segment 4), and VP6 (segment 9) which are in the core [24]. The non-structural proteins, NS1 (segment 5), NS2 (segment 8), NS3 (segment 10), and NS4 (segment 9) are located within infected cells and facilitate viral replication and release [20,22,25]. Non-structural protein 3 also facilitates the transport of the virus by acting as a bridge between VP2 and the host cell [22].

The segmented nature of the virus allows for reassortment, resulting in altered virulence and host ranges [17]. Howell et al. [26] identified seven serologically distinct serotypes and proposed naming them numerically based on the alphabetic order of the location in South Africa where the reference strain originated. Despite the suggestion to change the serotype nomenclature according to the date of virus isolation, the seven serotypes as suggested by Howell et al. [26] are prescribed by the International Committee on Taxonomy of viruses (ICTV) in the ICTV 9th Report of 2011 (https://talk.ictvonline.org/ictv-reports/ictv_9th_report/dsrna-viruses-2011/w/dsrna_viruses/188/reoviridae (accessed on 8 February 2021)). The seven EEV serotypes, based on the most variable protein, VP2, with the reference strains are: serotype 1 (EEV-1_Bryanston_RSA_1976), serotype 2 (EEV-2_Cascara_RSA_1967), serotype 3 (EEV-3_Gamil_RSA_1971), serotype 4 (EEV-4_Kaalplaas_RSA_1974), serotype 5 (EEV-5_Kyalami_RSA_1976), serotype 6 (EEV-6_Potchefstroom_RSA_1991), and serotype 7 (EEV-7_Northrand_RSA) [1,26].

The seven serotypes differ in prevalence under field conditions, probably due to variable adaptation to vectors, host and/or environment [15,26,27]. Isolation of virus from both clinical case material and field-collected *Culicoides* midges have shown the most common serotypes to be serotype 1, followed by serotype 6 [14,15,28]. In the past only limited numbers of clinical cases were described although a high seroprevalence of EEV in horses and donkeys in South Africa was shown [16,29]. This suggests that most EEV infections are subclinical in nature [30]. Although a variety of clinical signs, ranging from mild fever to encephalosis and death, have been ascribed to this disease [13,31], only a few cases have been laboratory confirmed through virus isolation, RT-PCR, or seroconversion [26]. The correlation between clinical signs and serotypes is lacking due to the limited genomic data available for EEV [32].

This study aims to describe the clinical presentation of horses that tested PCR positive for EEV submitted as part of arbovirus surveillance. Furthermore, we aimed to establish the molecular epidemiology of this virus in South Africa, through virus isolation and comparison of partial and complete genome sequences. 

## 2. Materials and Methods

### 2.1. Sample Collection

Samples were collected throughout South Africa as part of nationwide passive surveillance of horses to determine the etiological agents responsible for neurological, febrile, or respiratory signs or sudden unexplained death (SUD). Blood (EDTA or serum) and tissue specimens, such as brain, lung, and spleen, were submitted from across South Africa for pathogen identification. Epidemiological data including location and clinical signs of horses were submitted together with the samples. Clinical signs of sampled cases also included abortions, anorexia, orbital swelling, and icterus. All specimens were tested for flavi-, alpha-, and orthobunyaviruses as differential diagnoses at the Centre for Viral Zoonoses (CVZ) and for AHSV at the Equine Research Centre (ERC), Faculty of Veterinary Science, University of Pretoria. Results from both CVZ and ERC were combined to identify potential coinfection with other pathogens. 

*Culicoides* midge pools collected during the same period (2012–2017) from four sentinel surveillance sites in Johannesburg (Kyalami) and Pretoria (Boschkop) (Gauteng province), and Marekele National Park and Lapalala Wilderness (Limpopo province) were PCR screened for EEV as part of vector surveillance. 

### 2.2. RNA Extraction and Real-Time Reverse Transcription-PCR (rRT-PCR)

RNA was extracted from blood or tissue using the QIAmp Viral RNA mini kit (Qiagen, Valencia, CA, USA) or RNeasy mini kit (Qiagen, Valencia, CA, USA) respectively according to manufacturer’s instructions. Samples were tested for EEV using an in-house nested real-time hydrolysis probe RT-PCR, targeting the NS3 gene (Segment 10). In short, a 5× Transcriptor One-Step RT-PCR Kit (Roche, Basel, Switzerland) RT buffer, RT-PCR enzyme in combination with 20 pmol forward primer NS3F [33] and reverse primer NS3R (^799^5′-CCGAACTGGTACGGTA-3′^724^) were mixed with nuclease-free water to a total volume of 50 µL reaction following addition of 10 µL denatured (5 min at 95 °C) RNA. The reaction mix was subjected to initial incubation of 50 °C for 30 min, 94 °C for 7 min followed by 35 PCR cycles: 94 °C, 10 s; 52 °C, 30 min; 68 °C for 1 min supplemented by a final extension for 68 °C for 5 min. An EEV-specific nested real-time PCR using forward primer forward (NS3nF) (^198^5′-GGDGCRGATGARTGTGATAA-3′^217^) and reverse (NS3nR) (^616^5′-TTKCTAATYCTATCCGCGTTC-3′^596^) primer was performed using TaqMan™ assay (Roche, Basel, Switzerland) on the Roche LightCycler^®^ 2.0. Each reaction consisted of TaqMan™ Master Mix, 20 pmol per primer, 10 pmol of the EEV-specific FAM-TaqMan® probe (^498^5′-FAM-TGCGGTYTGATGARATGGAG-3′^517^) and 2 µL of first round product. Nuclease-free water was added to the reaction to give a total volume of 50 µL. Initial incubation was at 95 °C for 10 min followed by 45 PCR cycles: 95 °C for 10 s, 52 °C for 1 min, and 72 °C for 1 s, and cooled at 4 °C for 30 s.

### 2.3. Virus Isolation and Sequencing

All PCR-positive specimens were inoculated onto African green monkey kidney cells (Vero cells) for 7 days at 37 °C, 5% CO_2_ in Earle’s salts minimum essential medium containing L-glutamine, Mycozap™ Plus-CL and 2% fetal calf serum. Cultures were passaged three to four times and finally onto three confluent 75 cm^2^ flasks until 80% cytopathic effect (CPE) was observed.

Double-stranded RNA (dsRNA) was extracted from the infected cells of EEV cultures according to Potgieter et al. [34]. cDNA was synthesized using two methods. The first method included oligo-ligation with an anchor primer, PC3-T7 loop followed by sequence-independent cDNA synthesis and amplification (SISPA) [34,35,36]. Products were analyzed on a 1.5% agarose gel. The second method, rapid amplification of cDNA ends (RACE)-SISPA was performed on dsRNA according to Jansen van Vuren et al. [37] and Djikeng et al. [38]. cDNA was submitted to the Biotechnology Platform at the Agricultural Research Council (Pretoria, South Africa) for next-generation sequencing (NGS), using the Illumina TruSeq^®^ protocol of amplicon shearing and libraries preparation (Illumina, CA, USA). Sequencing was performed on an Illumina MiSeq (Illumina, CA, USA), generating ~600,000, 300 × 300 base pairs (bp) reads per sample.

The 418 bp EEV PCR-positive amplicons generated during the rRT-PCR were submitted for Sanger sequencing at Inqaba Biotec™ (Pretoria, South Africa) using a BigDye™ Direct Cycle Sequencing Kit (ThermoFisher Scientific, MA, USA). Sequence data generated of NS3 by Sanger sequencing or complete genomes using NGS were viewed and edited in CLC Genomic Workbench (version 8.0.1) (https://www.qiagenbioinformatics.com, accessed on 13 November 2019) and MEGA (version 6.06) (https://www.megasoftware.net, accessed on 11 November 2020). Using CLC Genomics, adapter sequences were trimmed from the NGS reads and de novo assembled into contigs. Contigs were mapped to reference EEV sequences obtained from GenBank and alignments were generated using all the EEV sequences generated in this study as well as from GenBank. Multiple sequence alignments were performed using the online version of MAFFT (version 7), (http://mafft.cbrc.jp/alignment/server/index.html, accessed on 11 November 2019) with default parameters. Maximum likelihood analyses were conducted in RAxML using GTR+G models of evolution [39]. Bootstrap support values were calculated using the autoMRE bootstopping criterion. P-distance analysis was calculated in MEGA 6.06 and used to determine nucleotide similarities between EEV samples and reference serotype strains. New sequences were submitted to GenBank and accession numbers were provided for the partial NS3 gene segment (MN921084-MN921120) as well as the full gene segments from six EEV isolates (MN956761-MN956704).

### 2.4. Statistical Analyses

All statistical analyses were done using Epi Info (https://www.cdc.gov/epiinfo/index.html, accessed on 8 February 2021). Fisher exact test was used to calculate odds ratio (OR) to determine possible associations (*p* < 0.05) between EEV-positive infections and clinical signs. Excluded from the OR calculations were animals with no clinical information (*n* = 37), animals that met sudden and unexplained death (SUD) (*n* = 9), and aborted fetuses (*n* = 11). The 95% confidence intervals (CI) were calculated where relevant.

## 3. Results

### 3.1. Epidemiological Analyses of Clinical Cases

From February 2010–December 2017, the Centre of Viral Zoonoses (CVZ) received blood and/or tissue specimens from 1523 horses presenting with undiagnosed neurological, febrile, and/or respiratory signs. Equine encephalosis virus RNA was PCR-detected in 111 horses (7.3%). Of the 111 EEV-infected horses, coinfections were detected in 17 (15.3%); i.e., 4/111 (3.6%) Middelburg virus (MIDV), 5/11 (4.5%) West Nile virus (WNV), and 8/111 (7.2%) AHSV.

Clinical signs significantly associated with EEV-positive cases in horses with recorded data (*n* = 106/1466) (OR > 1; *p* < 0.05) included pyrexia (OR 2.8 [1.8–4.3]), dyspnea (OR 5.7 [2.8–11.4]), and icterus (OR 2.2 [1.3–3.6]) (Table 1). In contrast, neurological signs (OR 0.2 [0.2–0.4]), specifically ataxia (OR 0.5 [0.3–0.8] and mortality (including euthanasia after presenting with clinical signs) (OR 0.3[0.2–0.6]), were inversely associated (OR < 1; *p* < 0.05) with EEV infections (Table 1). Although a high percentage of EEV-positive animals showed neurological abnormalities (47.7%) only 9.0% had a fatal outcome. Coinfections with MIDV, WNV, or AHSV were observed in horses with dyspnea (*n* = 2), pyrexia (*n* = 3), neurological signs (*n* = 8), and a combination of pyrexia and neurological signs (*n* = 4) with no deaths reported.

Positive cases were reported in eight of the nine South African provinces, with the highest number of EEV cases in the Northern Cape (16/110, 14.5%), followed by the Western Cape Province (30/366, 8.2%), and Gauteng (45/571, 7.9%) (Figure 1). No cases (0/18) were reported in Limpopo province. Annually, the average percentage of EEV-positive samples submitted to the CVZ between 2010 and 2017 was 6.3%. An increase in EEV infections was apparent in 2011 (26/170, 15.3%) and 2017 (47/423, 11.1%) (Figure 2). Equine encephalosis virus positivity was highest in April (38/267, 14.2%), followed by March (33/286, 1.5%), and February (12/148, 8.1%) (Figure 2). No EEV positives were detected in October or December.

### 3.2. Phylogenetic Analysis

A total of 51 EEV samples were isolated on cell culture, while dsRNA and complete cDNA were generated for only 21 samples using both cDNA synthesis methods. Of the 21, complete genome sequences were obtained for four EEV samples, SAE55/11, ZRU88/13/2, ZRU60/15, and ZRU148/17, while two samples, ZRU122/13/1, and ZRU083/15, had nine complete segments and a partial segment-4 (VP4) (Figure 3). Clinical information, location, isolation information, and genome lengths for each of the six EEV-positive horse sample are indicated in Table 2. The phylogenetic relatedness of all 10 segments was compared between the newly sequenced EEV samples, reference strains and four other EEV field isolates submitted to GenBank. Each of the segments was assigned a number or letter, based on its clustering with the seven reference serotype strains (Figure 3). The reference strains used were based on the classification determined by the sequences of segment 10 (accession numbers) as prescribed in the ICTV 9th Report of 2011 (EEV-1: FJ183393; EEV-2: AY115871; EEV-3: AY115874; EEV-4: AY115868; EEV-5: AY15869; EEV-6: AY115873; and EEV-7: AY115870). If two reference serotypes grouped together, they were assigned according to both serotypes, for example serotype 4 and 5 of segment 1, serotypes 1 and 4 of segment 3 and 4, as well as serotypes 1, 2, and 5, 7 for segment 10 (Figure 3). Field sample EEV-1_HS103_RSA_2006 formed a unique segment 8 (NS2) cluster (A), which included all six of the new isolates and EEV-1_88403_India_2008 (Figure 3). Similarly, the samples EEV-1_88403_India_2008 formed a unique cluster with two of the new isolates in segment 5 (NS1) and four samples in segment 7 (VP7) (Figure 3). It is evident that widespread reassortment occurred in EEV and additional sequencing of field isolates is essential. 

Isolates ZRU60/15 and ZRU83/15 had identical clustering across all 10 segments, with segments 1, 4, and 5 grouping with serotype 7, segments 3 and 9 with serotype 5, while segment 10 grouped with the serotypes 5 and 7 cluster (Figure 3). Similarly, ZRU88/13/4 and ZRU122/13/1 clustered with the same reference serotypes across all 10 segments. Segment 1 grouped with serotype 2, while segments 3, 4, and 5 grouped with serotype 7, segment 9 with serotype 5 and segment 10 with both serotypes 5 and 7 (Figure 3). In contrast, samples ZRU148_17 and SAE55_11 had unique clustering across the 10 genome segments (Figure 3).

Phylogenetic analysis on the VP2 segment indicated clustering of ZRU88/13/4, ZRU122/13/1, ZRU60/15, ZRU83/15, and ZRU148/17 with serotype 1 reference strain (EEV-1_Bryanston_RSA_1976) as well as field isolates EEV-1_HS103_RSA_2006 and forming a sister group with a previous EEV positive from India (EEV-1_88403_India_2008) (Figure 3). Sample SAE55/11 clustered with serotype 4 field isolate from Israel (EEV-4_Kimron_Israel), forming a sister group along the reference strain (EEV-4_Kaalplaas_RSA_1974) (Figure 3). Phylogenetic analysis of segment 6 (VP5) clustered ZRU88/13/4, ZRU122/13/1, ZRU60/15, and ZRU83/15 with serotype 1, while ZRU148/17 and SAE55/11 clustered with serotype 4 (Figure 3). 

Phylogenetic analysis of the partial NS3 gene region obtained from both horse and *Culicoides*-positive samples formed two distinct monophyletic groups (clade A and B) (bootstrap values (bs) = 100). The reference strains serotype 1, 2, and 4 clustered in clade A, while serotypes 3, 5, and 7 clustered within clade B (Figure 4). Results demonstrated close phylogenetic relationships with EEV-positive horses and EEVs detected in *Culicoides* midges during the same period. Based on the partial NS3 gene region, none of the horse or midge samples clustered with serotypes 3, 6, or the newly sequenced strain from India (accession number MG470867), while EEV-4_Kimron_Israel also grouped on its own in clade A (group C) (Figure 3 and Figure 4).

## 4. Discussion

Equine encephalosis virus was detected in eight provinces of South Africa from 2010–2017 with pyrexia, respiratory signs, and icterus significantly associated with EEV. In contrast, neurological signs and mortality were inversely associated with EEV infections, despite a high number (48.1%) of cases reporting neurological signs with mortality higher (10/111, 9.0%) than previously reported [32,40]. The average age of horses that died was lower than previously reported at 5.1 years, with the youngest being 4 months and the oldest 12 years [5]. Some of the horses in this study were euthanized for humane reasons which could bias estimations. 

The incidence of EEV infections detected at CVZ was highest in 2011 in the Western Cape Province (24.5%), which coincides with an AHSV outbreak in the same province [18]. This could be due to an increase in surveillance and diagnostic submissions of specifically horses displaying neurological or other non-typical AHS symptom, while typical AHS cases were submitted to either the ERC or Agricultural Research Council–Onderstepoort Veterinary Institute. Therefore, the number of cases received at CVZ would not have reflected the total incidence of EEV during those years. Equine encephalosis and AHS usually present different symptoms, although there may be cases where it is not possible to distinguish clinically between them, complicating diagnoses, especially during coinfections of both viruses [41]. The high incidence of EEV in 2017 was due to outbreaks occurring in Gauteng, Western Cape and KwaZulu-Natal provinces. Simultaneously, an increase in outbreaks of Middelburg and West Nile viruses were reported, possibly due to warmer and wetter conditions following drought, resulting in an increase in vector populations (to be reported elsewhere).

These data analyses showed that differential diagnostic testing is essential to resolve outbreaks and to investigate virus circulation of not just World Organization for Animal Health (OIE) notifiable viruses such as AHSV but also to detect other emerging economically important viruses such as EEV [5,32]. The data analyses also suggested that EEV may contribute to the morbidity and mortality of horses in South Africa despite the lower fatality rate relative to AHSV. A limitation of the study remains that all other possible infectious and non-infectious etiologies could not be excluded by comprehensive investigations in all cases, suggesting the causative link with clinical signs still has to be regarded with caution.

Full genome sequencing of six isolates submitted between 2011 and 2017 identified the isolates as serotype 1 and serotype 4 based on nucleotide and amino acid similarities of the outer capsid proteins, VP2 and VP5. This indicates that these two serotypes might be the dominant serotypes circulating in South Africa currently and not serotype 1 and 6, as previously reported [14,15,30]. Based on phylogenetic and p-distance analyses of nucleotide and amino acid similarities, the full NS3 segment of newly sequenced strains also clustered according to serotypes, serotypes 4 or a serotypes 5 and 7 cluster. Phylogenetic analysis of the partial NS3 (332 bp) gene region showed no geographical grouping of EEV in midges with the majority grouping with serotypes 4 and 7. Geographical and syndromic grouping, based on the partial NS3 gene, were evident for EEV-positive horses with respiratory signs and horses from the Northern Cape province all clustering within clade A, serotype 4 (bs = 82). Non-structural protein 3 is involved in the viral replication and viral release [6,25], which could explain the close relationship between EEV strains detected in midge pools and horses. The phylogram obtained from the full NS3 gene region clustered serotype 7 with serotypes 3, 5, and 6 into clade B rather than with serotypes 1, 2, and 4 into clade A [3,33]. The difference could be due to different reference serotypes 7 strains used, as Van Niekerk et al. [33] and Aharonson-Raz et al. [3] used a field isolate from KwaZulu-Natal whereas this study used the Northrand (EEV-7_Northrand_RSA_2008) isolate from Gauteng. Van Niekerk et al. [33] were the first to investigate the genetic relationship of the seven EEV serotypes and theirs is therefore still the nomenclature used by the ICTV (https://talk.ictvonline.org/ictv-reports/ictv_9th_report/dsrna-viruses-2011/w/dsrna_viruses/188/reoviridae accessed on 8 February 2021). It should however be noted that the numbering of the serotypes in the Yadav et al. [4] study was assigned based on suggestions made to the ICTV VIIIth Report, where Cascara is serotype 1, Gamil serotype 2, Kaalplaas serotype 3, and Bryanston serotype 4 [1,26]. Reclassification of orbivirus serotypes based on year of isolation rather than published nomenclature could be confusing to our current understanding of AHSV and BTV, where the oldest isolates are respectively AHSV-1_1180_1933 followed by AHSV-5_Westerman_1936 and BTV-8_Camp_1937 followed by BTV-12_Byenespoort_1941.

Based on the full gene segments of all 10 genome segments, widespread reassortment occurred through all the field-isolated EEV samples. Sample SEA55/11, the only sample identified as serotype 4 based on segment 2 (VP2), was the only sample that originated from a horse in the Northern Cape as well as the only horse that presented with dyspnea. This could possibly indicate difference in serotypes by geographical location and/or clinical signs as demonstrated for the partial NS3 gene, although more samples from this region and from horses with dyspnea should be included. All three horses (ZRU122/13/1, ZRU60/15, and ZRU83/15) that presented with neurological disease were classified as serotype 1 based on segment 2 (VP2), with a segment 10 (NS3) from serotypes 5 and 7. Genetic analysis in this study suggested that although the identified strains clustered with the original reference strains, several had drifted significantly and had as little as 81.6% amino acid similarity to the serotypes they clustered with when analyzing segment 2. The limited similarity of the identified strains in several proteins to the existing serotype reference strains suggests genetic drift and likely resulted in immune evasion. Future work including extensive bioinformatics reviewing of available EEV full genomes should be conducted to determine whether the NS4 gene is present in EEV [20].

A positive EEV culture from sample ZRU148/17 was obtained from placental tissue from the fetus taken from the deceased mare. Both spleen and lung tissue of the mare tested positive for EEV by PCR, demonstrating possible cross-placental transmission of EEV. Vertical placental transmission of BTV has been shown [42,43]; however, no evidence for vertical transmission of either EEV or AHSV has been reported before [44]. It is important to note that abortion during early gestation (first 5 to 6 months) has previously been linked to EEV-infected mares [7] and that previous work done on maternal antibodies and EEV infection reported that EEV maternal antibodies may not prevent EEV infection of foals [31].

Four of the 21 cultures sequenced had complete AHSV genomes, although they were originally PCR-positive for EEV NS3. This could indicate coinfection of EEV and AHSV in the four horses, with AHSV outcompeting EEV during cell culture isolation. Equine encephalosis virus also has similar characteristics in cell culture to AHSV virus, making differentiation based on CPE impossible [45]. This collection of EEV isolates increased the number isolated in South Africa and demonstrated active infection of EEV. Previous studies have concluded that the limited number of isolates of EEV in South Africa proved subclinical infections [30]. This collection would allow for further studies into the mechanisms of infection and vaccine development. Isolation of all EEVs from Vero cells also demonstrates the possibility of using cell lines other than BHK, which is usually used in South Africa for EEV isolations.

This study suggests that EEV may be significantly associated with signs of pyrexia, dyspnea, and icterus, while neurological signs did not necessarily predict an EEV infection, although 47% of cases did have neurological signs, with ataxia (25.2%) and paresis (14%) being most frequently recorded. Equine encephalosis virus seems to be more prevalent than AHSV in South Africa [23], suggesting additional factors might be responsible for EEV transmission such as vectors [16], the lack of vaccination against EEV and/or the highly variable nature of EEV. Antibodies against EEV have been noted in zebras [27,41], donkeys [7,41] and African elephants [46], indicating possible reservoir hosts for the virus.

This study describes the clinical epidemiology of EEV cases identified over seven years in South Africa. It demonstrated a high incidence of EEV infection in horses with clinical signs in South Africa, suggesting the virus may contribute significantly to the morbidity in these animals and may be more important than previously recognized. We further highlight the high reassortment ability of the virus. Clinical cases were associated with serotype 1 and 4 based on VP2 in horses in South Africa, but other genome segments clustered with several other serotypes that may be important in outbreak scenarios and differences in disease severity.

## Figures and Tables

**Figure 1 viruses-13-00398-f001:**
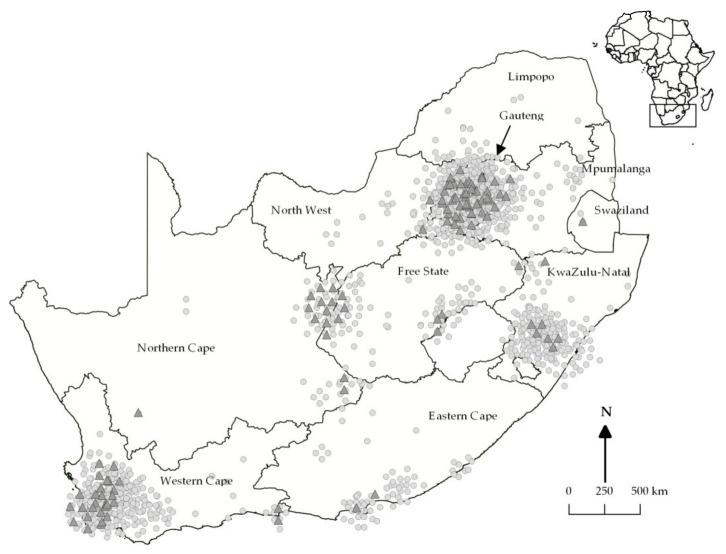
Locations of equine encephalosis virus PCR-positive (triangles) and -negative (circles) samples from horses, South Africa, 2010–2017. Inset shows location of South Africa in Africa.

**Figure 2 viruses-13-00398-f002:**
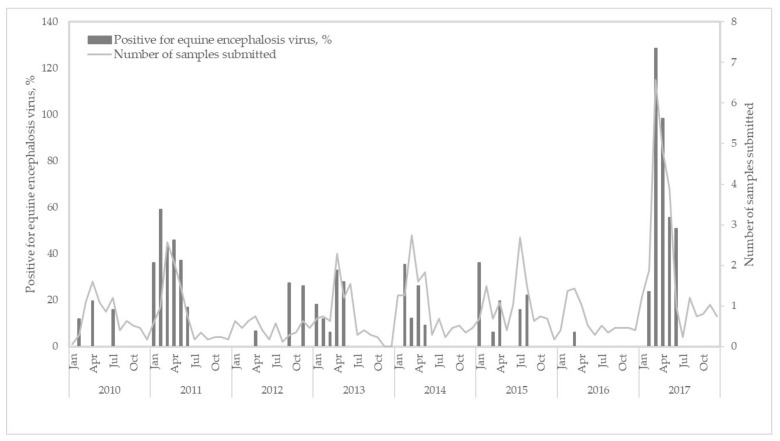
Seasonal detection of equine encephalosis virus-positive infections in horses in South Africa, 2010–2017. The left axis indicates the EEV positivity and the right axis the number of samples submitted.

**Figure 3 viruses-13-00398-f003:**
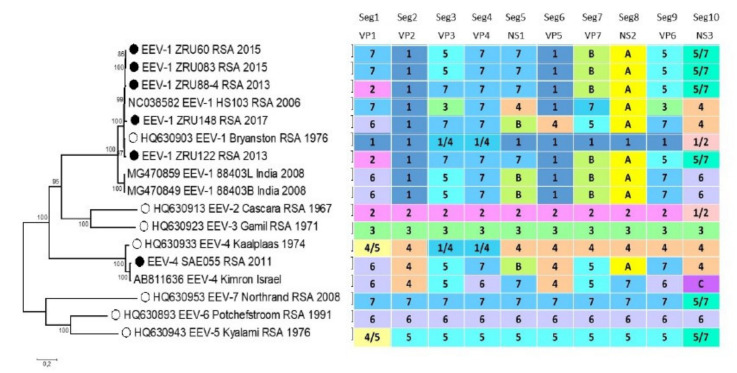
Maximum likelihood phylogram of the complete gene segment 2 of EEV (3206 nucleotides, taxa = 17, model: GTR+G, rooted at mid-point). Black circles indicate newly sequenced EEV isolates (Accession number: MN956740–MN956745), while white circles represent the reference strains. Clustering of all ten segments is indicated next to each sample. These are colored according to the reference serotype or field isolate group assigned with each segment.

**Figure 4 viruses-13-00398-f004:**
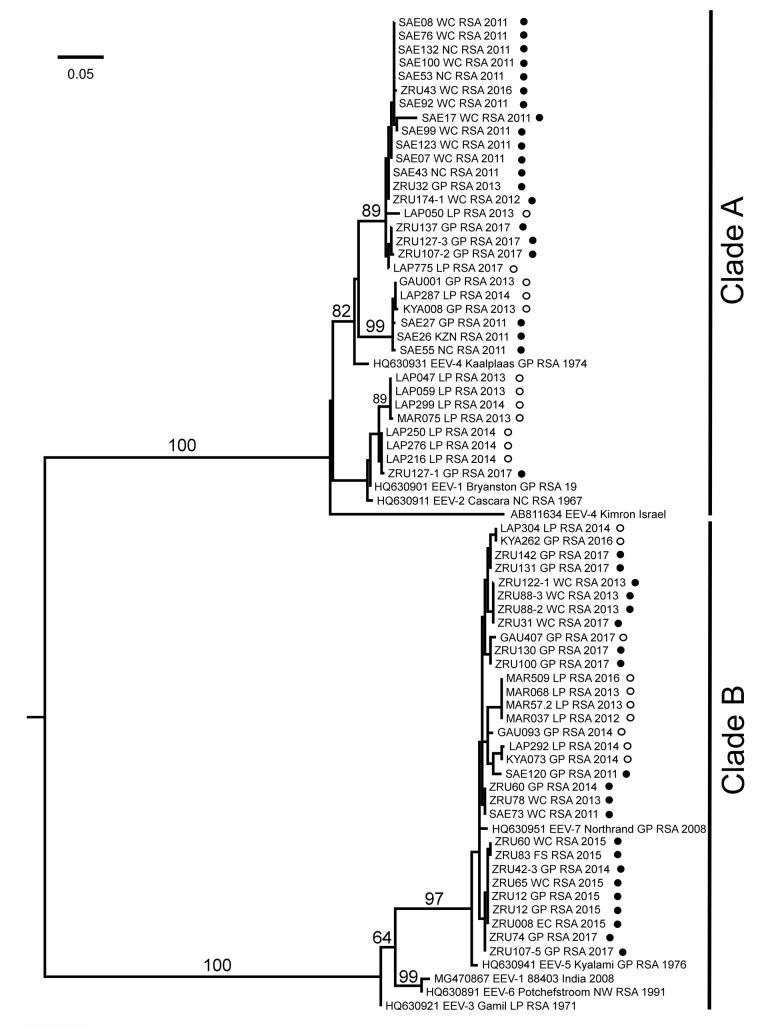
Maximum likelihood phylogram of the partial NS3 gene (332 nucleotides, taxa = 73, model: GTR+G, rooted at mid-point) of equine encephalosis virus strains detected in horses (filled circles) (accession numbers MN921084-MN921120) and midges (open circles) (accession numbers MN271001-MN271022). Sample names consist of the province from which the sample originated (GP: Gauteng, WC: Western Cape; NC: Northern Cape; EC: Eastern Cape; LP: Limpopo; KZN: KwaZulu-Natal; NW: North West FS: Free State), country (RSA: South Africa) and year of collection.

**Table 1 viruses-13-00398-t001:** Clinical signs reported in horses upon submission of samples to the Centre for Viral Zoonoses from 2010–2017. Clinical signs significantly associated with equine encephalosis virus-positive animals are indicated in bold. Excluded from the odds ratio (OR) calculations were animals with no clinical information (*n* = 57).

Clinical Sign	EEV Positive (*n* = 106; %)	EEV Negative (*n* = 1360; %)	OR [95.0% Confidence Interval (CI)]	*p*-Value
Pyrexia	77 (72.6%)	664 (48.8%)	2.8 [1.8–4.3]	0.00
Icterus	20 (18.8%)	132 (9.7%)	2.2 [1.3–3.6]	0.00
Dyspnea	12 (11.3%)	30 (2.2%)	5.7 [2.8–11.4]	0.00
Neurological	51 (48.1%)	1093 (80.4%)	0.2 [0.2–0.3]	0.00
Ataxia	27 (25.5%)	535 (39.3.%)	0.5 [0.3–0.8]	0.00
Recumbency	9 (8.5%)	211 (15.5%)	0.5 [0.3–1.0]	0.06
Paralysis	6 (5.7%)	127 (9.3%)	0.6 [0.3–1.4]	0.30
Paresis	15 (14.1%)	189 (13.9%)	1.0 [0.6–1.8]	0.90
Depression	7 (6.6%)	72 (5.3%)	1.3 [0.6–2.8]	0.51
Supra orbital swelling	2 (1.9%)	29 (2.1%)	0.9 [0.2–3.8]	1.00
Swollen limbs	1 (0.9%)	21 (1.5%)	0.6 [0.1–4.6]	1.00
Paddling	0	20 (1.5%)	Undefined	0.39
Outcome	*N* = 111	*N* =1412	OR [95.0% CI]	*P*-value
SUD	0	9 (0.6%)	Undefined	1.0
Abortion	1 (0.9%)	10 (0.7%)	1.3 [0.2–10.0]	0.57
Fatal	11 (9.9%)	358 (25.2%)	0.3 [0.2–0.6]	0.00
No clinical history	4 (3.6%)	33 (2.3%)	1.5 [0.5–4.3]	0.4

SUD: sudden unexplained death.

**Table 2 viruses-13-00398-t002:** Equine encephalosis virus isolates from which full gene segments were obtained indicating the submission date, associated clinical sign, location, and sample type.

Sample Code	Month Year	Main Sign Upon Submission	Province	Specimen Isolate Obtained From	Isolate (Vero) Passage Number	VP2 SeroType
SAE055/11	Mar-11	Dyspnea	Northern Cape	EDTA	5 (93)	4
ZRU088/13/2	Apr-13	Fever	Western Cape	EDTA	5 (95)	1
ZRU122/13/1	May-13	Neurological	Western Cape	EDTA	5 (95)	1
ZRU060/15	Mar-15	Neurological	Western Cape	EDTA	5 (95)	1
ZRU083/15	Apr-15	Neurological	Free State	EDTA	5 (95)	1
ZRU148/17	Mar-17	Abortion	North West	Fetal Placenta	4 (95)	1

## Data Availability

The data that support the findings of this study are openly available in GenBank, accession numbers MN921084–MN921120 and MN956761–MN956704.

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
