# Peer review of "Epidemiology and Genomic Analysis of Equine Encephalosis Virus Detected in Horses with Clinical Signs in South Africa, 2010–2017"

_viruses, 2021, doi:10.3390/v13030398_

Round 1
Reviewer 1 Report
This is interesting work but is sadly let down by poor manuscript editing and consistency- It also requires a better structure and tighter wording to enhance the readers grasp of the disease and importance of the work. There are several inconsistencies throughout the results hence clearer presentation of numbers required -I suggest the authors seek to re write the introduction and discussion perhaps with guidance form an external experienced manuscript writer
There are many comments however I believe the whole manuscript needs to be more informative relating to EEV with more clarity and concision
all the best
some detailed suggestions attached

Author Response
Thank you for the valuable comments, please see below each comment followed by our response.
Equine encephalosis virus in horses in South Africa: Clinical signs and molecular epidemiology
Suggest change to reflect work
Detection of Equine encephalosis virus and molecular epidemiology in horses with clinical signs of disease and post mortem samples in South Africa
- Agree to change the title to reflect the work that was done as follows: Epidemiology and genomic analysis of equine encephalosis viruses detected in horses with clinical signs in South Africa, 2010-2017
This is interesting work but is sadly let down by poor manuscript editing and consistency- It also requires a better structure and tighter wording to enhance the readers grasp of the disease and importance of the work -I suggest the authors seek to re write the introduction and discussion perhaps with guidance form an external experienced manuscript writer
There are many comments however I believe the whole manuscript needs to be more informative relating to EEV and concise
Abstract
Suggest replace
Equine encephalosis virus (EEV) is a neglected virus endemic to South Africa, regarded 22 as only causing mild infections in equines. Specimens from horses that presented with undefined neurological, febrile, or respiratory signs, or sudden and unexpected death
With
Equine encephalosis virus (EEV) is a virus endemic to South Africa, that is considered to generally result in mild disease signs following infection of horses in most cases. The virus and clinical consequences of infections in horses however remains poorly understood, as only causing mild. Specimens were analysed from live horses that presented with neurological, febrile, or respiratory signs of unknown
- I have changed this part to. “Equine encephalosis virus (EEV) is a neglected virus endemic to South Africa and is considered to generally result in mild disease in equines. Specimens were analyzed from live horses that presented with undefined neurological, febrile, or respiratory signs, or sudden and unexpected death.”
Line 26 ?? exactly how many were EEV positive 119 or 107 be consistent or explain and use both % and numerator denominator to make clearer please
111 of 1529 (7.3%)
- For the statistical analyses only horses that were documented as having clinical signs were included in the analyses. The horses that had no clinical information (EEV +: n=4; EEV -: n=52) upon submission were excluded from analyses. Therefore, 111 horses tested positive for EEV but 4 were submitted without any clinical information even though we tried to get this information. It is also stated under statistical analyses “Excluded from the OR calculations were animals with no clinical information (n=37), animals that suddenly and unexpectedly died (SUD) (n=9) or aborted fetuses (n=10). The 95% Confidence Intervals (CI) were calculated where relevant.”
Seventeen of the EEV positive horses (%) were detected to be coinfected with other viruses including ?/ ?, (4.5%),? WNV Nile- ?/ ?, (3.5%),? Middelburg- (3.5%) or African Horse sickness virus (?/ ?, 7.2%).
- Changed to: “Seventeen of the EEV positive horses also had co-infections with either West Nile- (5/107, 4.5%), Middelburg- (4/107, 3.5%) or African Horse sickness virus (8/107, 7.2%).
Line 35 replace were observed. With was observed
- Changes made
Introduction
This needs to be reworded and made more succinct and clearly introduce the topics of interest
Reword whole section after the first sentence to properly describe (or introduce) EEV the virus, transmission, the ranges of disease and diagnosis in an orderly fashion
Replace or re phrase the first paragraph please to properly describe EEV -currently there is more information about AHS!
- The introduction was extensively changed as to introduce EEV
Line 44
studies on the disease and virus is lacking
replace with studies on the disease and virus are lacking
line 50 replace of the disease with
of disease delete (the)
- Changes made
line 51 replace numerous epizoonotics in the Mediterranean “with numerous epizootics in the Mediterranean”
- Changes made
I would reorder these sentences for clarity line 51- ……This raises the concern of 52 emergence of these orbiviruses in areas where the Culicoides vector occurs. The vectors for EEV, hematophagous biting midges in the genus Culicoides, (13-16) are also an important 54 vector for other economically important arboviruses such as BTV and AHSV
*To perhaps. The vectors for 53 EEV, hematophagous biting midges in the genus Culicoides, (13-16) are also an important 54 vector for other economically important arboviruses such as BTV and AHSV.
line 52-54 needs to be reorganized to introduce Culicoides to readers please
- *Changes to “Similar to other orbiviruses, EEV is a non-infectious arbovirus transmitted by hematophagous biting midges in the genus Culicoides (Diptera: Ceratopogonidae) (13-16). The predominant EEV associated Culicoides species- Culicoides imicola and Culicoides Bolitinos- are also important vectors for other economically important arboviruses such as BTV and AHSV
Not so much detail re serotypes in Introduction needs to be brief and succinct -consider moving some sections to discussion for expanded serotypes section
- Currently there are 2 ways of numbering the serotypes so it is crucial to say what numbering we are using early on.
Line 81 The seven serotypes are not equally prevalent under field conditions probably due
Better to say
The Serotypes differ in prevalence the
- Changed to “The seven serotypes differ in prevalence under field conditions probably due to variable adaptation to vectors, host and/or environment”
Line 92---change to
This study aims to describe the clinical presentation of horses that tested PCR positive for EEV in horses with neurological disease. Furthermore we aimed establish the molecular epidemiology of this virus in South Africa, through virus isolation and comparison of partial or complete genomes sequences.
- Changed to “This study aims to describe the clinical presentation of horses that tested PCR positive for EEV submitted as part of arbovirus surveillance. Furthermore, we aimed to establish the molecular epidemiology of this virus in South Africa, through virus isolation and comparison of partial and complete genome sequences. “
Statistical analyses of clinical cases
Table 1 is useful however
Need to check table headings for % figures ie 107/1529 7.3% or 111/1529 ???
- As explained above, only clinical cases included. Under “Outcome” all cases were included
Then please ensure consistency with text- I would state actual P values please of findings and SUD? in legend please
- I added a footnote explaining Sudden unexplained deaths (SUD). These animals were also left out of the calculation to determine clinical association. Actual p-values were added
Replace line 174 needs to be clearer and better arranged
Of these, four horses (3.6%) had a co-infection with Middelburg virus (MIDV), five 174 with West Nile virus (WNV) (4.5%) and eight with AHSV (7.2%).
111/1529 horses 7.2 %
Eg Of the 111 EEV infected horses coinfections were detected in 17 (15.3%)
- Changed to: “Of the 111 EEV infected horses co-infections were detected in 17 (15.3%) e. 4/111 (3.6%) Middelburg virus (MIDV), 5/11 (4.5%) West Nile virus (WNV) and 8/111 (7.2%) AHSV (7.2%).”
Discussion
From 2010-2017, es originating in eight provinces of South Africa. relative to EEV negative cases.
Suggest change to a tighter summary of results
The presence of EEV was detected in horses from South Africa with clinical signs of illness including Pyrexia, respiratory signs, and icterus were significantly associated with EEV
- Edited the sentence “Equine encephalosis virus was detected in eight provinces of South Africa from 2010-2017 with pyrexia, respiratory signs, and icterus significantly associated with EEV.”
Need to discuss possibility of detection in non clinical horses??? And possibility of other agents not detected having a role - given the fact that many EEV infected horses may show minimal clinical SIGNS
It is important to discuss that the finding of EEV may be incidental in many cases and that the association may or may not be causation -possibly even a co factor to produce associated disease
- Added “A limitation of the study remains that all other possible infectious and non-infectious etiologies could not be excluded by comprehensive investigations in all cases suggesting the causative link with clinical signs still has to be regarded with caution.”
please discuss this
I would suggest for line 350 additional factors might be responsible for EEV infection-such as vectors etc.,
- Changed to: “Equine encephalosis virus seems to be more prevalent than AHSV in South Africa (23) suggesting additional factors might be responsible for EEV transmission such as vectors (16), the lack of vaccination against EEV”
Concluding paragraph should read far more clearly -and summarise overall findings leaving that detail earlier in discussion re WNV AHSV and confusing serotype association descriptions
- Agree, changed to: “This study describes the clinical epidemiology of EEV cases identified over 7 years in South Africa. It demonstrated a high incidence of EEV infection in horses with clinical signs in South Africa, suggesting the virus may contribute significantly to the morbidity in these animals and may be more important than previously recognized. We further highlight the high reassortment ability of the virus. Clinical cases were associated with serotype 1 and 4 based on VP2 in horses in South Africa but other genome segments clustered with several other serotypes that may be important in outbreak scenarios and differences in disease severity.
Reviewer 2 Report
The title is misleading and must be changed. The words “clinical signs” must be removed from the title, since
- the manuscript is not primarily about the clinical signs, but the focus is on molecular epidemiology
- the authors themselves did not see any of the horses to testify to the clinical observations, they only received the blood and tissue samples from a variety of collaborators/individuals around the country
line 88 -- replace “encephalitis” with “encephalosis” as documented in Erasmus et al 1970; encephalitis was never documented for EEV
The issue of the numbers/names of the EEV serotypes
lines 34, 76-80, 307-309, etc -- all lines, tables, and figures wherein EEV serotypes are indicated by numbers must be corrected.
The numbering of the serotypes as referred to in lines 308-310 is correct and not “erroneously” as stated in line 308.
In virology the convention is that viruses and their related isolates/serotypes/strains are named chronologically based on date of isolation and not alphabetically based on names. In the case of EEV, one publication, which is the one the manuscript uses as a reference, is responsible for the tremendous confusion in EEV taxonomy (Howell PG, Groenewald D, Visage CW, Bosman AM, Coetzer JA, Guthrie AJ (2002) The classification of seven serotypes of equine encephalosis virus and the prevalence of homologous antibody in horses in South Africa. Onderstepoort J Vet Res 69:79–93).
The authors should consult
- https://www.reoviridae.org/dsRNA_virus_proteins/ReoID/EEV-isolates.htm
- Mertens PP, Maan S, Samuel A, Attoui H. Orbivirus, Reoviridae. In: Fauquet CM, Mayo MA, Maniloff J, Desselberger U, Ball LA, editors. Virus taxonomy, VIIIth report of the ICTV. London: Elsevier/ Academic Press; 2005. p. 466–83
- Quantitative RT-PCR assays for identification and typing of the Equine encephalosis virus. Maan S, Belaganahalli MN, Maan NS, Potgieter AC, Mertens PPC. Braz J Microbiol. 2019 Jan;50(1):287-296. doi: 10.1007/s42770-018-0034-1. Epub 2018 Dec 10. PMID: 30637652
References – - The original references must be used. All the references in the manuscript must be scrutinised and replaced with the correct original and applicable references that contain the data/facts mentioned. Many references in the manuscripts are 3rd hand - references that refer to references that refer to the original refence.
For example,
reference 1 should be replaced by
- Theiler A. Notes on a fever in horses simulating horse-sickness. Transvaal Agricultural Journal. 1910; 8:581–6.
- Erasmus BJ, Adelaar TF, Smit JD, Lecatsas G, Toms T. The isolation and characterization of equine encephalosis virus. Bull Off Int Epizoot. 1970; 74:781–9.
reference 3 should be replaced by
- Isolation and phylogenetic grouping of equine encephalosis virus in Israel. Aharonson-Raz K, Steinman A, Bumbarov V, Maan S, Maan NS, Nomikou K, Batten C, Potgieter C, Gottlieb Y, Mertens P, Klement E.Emerg Infect Dis. 2011 Oct;17(10):1883-6. doi: 10.3201/eid1710.110350.PMID: 22000361
References 7 and 15
are not applicable to the statement in lines 76-80
and many more reference problems
Author Response
Thank you for the valuable comments, please see below each comment followed by our response.
The title is misleading and must be changed. The words “clinical signs” must be removed from the title, since
- the manuscript is not primarily about the clinical signs, but the focus is on molecular epidemiology
- the authors themselves did not see any of the horses to testify to the clinical observations, they only received the blood and tissue samples from a variety of collaborators/individuals around the country
line 88 -- replace “encephalitis” with “encephalosis” as documented in Erasmus et al 1970; encephalitis was never documented for EEV
- The title has been changed to “Epidemiology and genomic analysis of equine encephalosis virus detected in horses with clinical signs in South Africa, 2010-2017”
The issue of the numbers/names of the EEV serotypes
lines 34, 76-80, 307-309, etc -- all lines, tables, and figures wherein EEV serotypes are indicated by numbers must be corrected.
The numbering of the serotypes as referred to in lines 308-310 is correct and not “erroneously” as stated in line 308.
In virology the convention is that viruses and their related isolates/serotypes/strains are named chronologically based on date of isolation and not alphabetically based on names. In the case of EEV, one publication, which is the one the manuscript uses as a reference, is responsible for the tremendous confusion in EEV taxonomy (Howell PG, Groenewald D, Visage CW, Bosman AM, Coetzer JA, Guthrie AJ (2002) The classification of seven serotypes of equine encephalosis virus and the prevalence of homologous antibody in horses in South Africa. Onderstepoort J Vet Res 69:79–93).
The authors should consult
- https://www.reoviridae.org/dsRNA_virus_proteins/ReoID/EEV-isolates.htm
- Mertens PP, Maan S, Samuel A, Attoui H. Orbivirus, Reoviridae. In: Fauquet CM, Mayo MA, Maniloff J, Desselberger U, Ball LA, editors. Virus taxonomy, VIIIth report of the ICTV. London: Elsevier/ Academic Press; 2005. p. 466–83
- Quantitative RT-PCR assays for identification and typing of the Equine encephalosis virus. Maan S, Belaganahalli MN, Maan NS, Potgieter AC, Mertens PPC. Braz J Microbiol. 2019 Jan;50(1):287-296. doi: 10.1007/s42770-018-0034-1. Epub 2018 Dec 10. PMID: 30637652
- We used the serotype references as stipulated by the International Committee on Taxonomy of Viruses (ICTV), Report 9 of 2011. The serotype classification there is still according to Howell et al., 2002 (https://talk.ictvonline.org/ictv-reports/ictv_9th_report/dsrna-viruses-2011/w/dsrna_viruses/188/reoviridae).
- Reclassification of orbivirus serotypes based on year of isolation rather than published nomenclature could be confusing to our current understanding of AHSV and BTV, where the oldest isolates are respectively AHSV-1_1180_1933 followed by AHSV-5_Westerman_1936 and BTV-8_Camp_1937 followed by BTV-12_Byenespoort_1941.
References – - The original references must be used. All the references in the manuscript must be scrutinised and replaced with the correct original and applicable references that contain the data/facts mentioned. Many references in the manuscripts are 3rd hand - references that refer to references that refer to the original refence.
For example,
reference 1 should be replaced by
- Theiler A. Notes on a fever in horses simulating horse-sickness. Transvaal Agricultural Journal. 1910; 8:581–6.
The article by Theiler did not confirm the disease as EEV. Rather used the Erasmus et al 1970 article
- Erasmus BJ, Adelaar TF, Smit JD, Lecatsas G, Toms T. The isolation and characterization of equine encephalosis virus. Bull Off Int Epizoot. 1970; 74:781–9.
- Changes made
reference 3 should be replaced by
- Isolation and phylogenetic grouping of equine encephalosis virus in Israel. Aharonson-Raz K, Steinman A, Bumbarov V, Maan S, Maan NS, Nomikou K, Batten C, Potgieter C, Gottlieb Y, Mertens P, Klement E. Emerg Infect Dis. 2011 Oct;17(10):1883-6. doi: 10.3201/eid1710.110350.PMID: 22000361
- Changes made throughout the manuscript
References 7 and 15
- The references were deleted
are not applicable to the statement in lines 76-80
and many more reference problems
- References were checked and edited where applicable.
Reviewer 3 Report
The manuscript entitled "Equine encephalosis virus in horses in South Africa: Clinical signs and molecular epidemiology" is an excellent and fine analysis of the molecular epidemiology of equine encephalosis virus in South Africa. Clinical surveillance in horses and active collection of Culicoides midges allowed the authors to collect a large diversity of EEV isolates.
Several comments should be considered before publication :
Line 41 : Reoviridae and orbivirus should be in italics
Lines 53-55 : Could the authors provide a brief overview of Culicoides species involved in EEV transmission ?
Lines 75-79 : EEV strains have been classified differently by other consortia, see for example https://www.reoviridae.org/dsRNA_virus_proteins/ReoID/EEV-isolates.htm
With EEV-1 corresponding to the Cascara strain
EEV-2 corresponding to the Gamil strain
EEV-3 corresponding to the prototypic Kaalplas strain
EEV-4 corresponding to the Bryanston strain
Could the authors provide updated information on strain classification?
Lines 116-125 : why were real-time PCR products analysed in gel electrophoresis ? why was a nested PCR used, was the real-time RT-PCR alone not sensitive enough to detect EEV ? A 418bp-amplicon should be suboptimally amplified in real-time PCR (classically, amplicons lengths are 150-200nt max). How was the risk of contamination, associated with nested PCRs, controled ? If the cycling conditions have not been described elsewhere, please detail them in the manuscript. Also indicate which fluorochroms were bound to the TaqMan probe.
Table 1, why are there only 107 positive EEV cases considered in the table, while 111 positive animals were reported L173. What does « SUD » means ? Horses with neurological symptoms are overrepresented in the cohort, could the authors discuss the bias originating from it during data analysis and the confidence about the prevalence of neurological signs in EEV positive animals.
Lines 185-189 : would be interesting here to draw a map and a temporal analysis of EEV cases in South Africa.
Figure 2 is incomplete (clades A and B missing). It would be interesting here to indicate which EEV serotypes have been evidenced in horses and mosquitoes and from which region ; a more detailed analysis would allow to map the diversity of EEV strains in the different regions of South Africa.
Author Response
Thank you for the valuable comments, please see below each comment followed by our response.
The manuscript entitled "Equine encephalosis virus in horses in South Africa: Clinical signs and molecular epidemiology" is an excellent and fine analysis of the molecular epidemiology of equine encephalosis virus in South Africa. Clinical surveillance in horses and active collection of Culicoides midges allowed the authors to collect a large diversity of EEV isolates.
Several comments should be considered before publication :
Line 41 : Reoviridae and orbivirus should be in italics
- Changes made.
Lines 53-55 : Could the authors provide a brief overview of Culicoides species involved in EEV transmission ?
- Changes made.
Lines 75-79 : EEV strains have been classified differently by other consortia, see for example https://www.reoviridae.org/dsRNA_virus_proteins/ReoID/EEV-isolates.htm
With EEV-1 corresponding to the Cascara strain
EEV-2 corresponding to the Gamil strain
EEV-3 corresponding to the prototypic Kaalplas strain
EEV-4 corresponding to the Bryanston strain
Could the authors provide updated information on strain classification?
- We used the serotype references as stipulated by the International Committee on Taxonomy of Viruses (ICTV), Report 9 of 2011. The serotype classification there is still according to Howell et al., 2002 (https://talk.ictvonline.org/ictv-reports/ictv_9th_report/dsrna-viruses-2011/w/dsrna_viruses/188/reoviridae)
- Reclassification of orbivirus serotypes based on year of isolation rather than published nomenclature could be confusing to our current understanding of AHSV and BTV, where the oldest isolates are respectively AHSV-1_1180_1933 followed by AHSV-5_Westerman_1936 and BTV-8_Camp_1937 followed by BTV-12_Byenespoort_1941.
Lines 116-125 : why were real-time PCR products analysed in gel electrophoresis ? why was a nested PCR used, was the real-time RT-PCR alone not sensitive enough to detect EEV ? A 418bp-amplicon should be suboptimally amplified in real-time PCR (classically, amplicons lengths are 150-200nt max). How was the risk of contamination, associated with nested PCRs, controled ? If the cycling conditions have not been described elsewhere, please detail them in the manuscript. Also indicate which fluorochroms were bound to the TaqMan probe.
- We analyzed the real-time PCR with gel electrophoresis to excise the amplicon and sequence to use in phylogenetic analyses (Figure 2). Also, a benefit of optimizing a larger amplicon length when using real-time PCR.
- The PCR we used in the lab has been extensively optimized to detect EEV in various types of samples and is the standard PCR we use in the lab. Some samples such as midges do have a low viral load and the added sensitivity to the nested PCR allows us to detect more positives that would have been lost.
- As for contamination, the laboratory is set up as a diagnostic laboratory, so we use different laboratories for each step as well as non-template controls for each step and the positives controls and samples are always added separate. We also compare the sequences with our positive control.
- I have included more details in the PCR method including the fluorochroms: “In short, a 5x Transcriptor One-Step RT-PCR Kit (Roche, Basel, Switzerland) RT buff-er, RT-PCR enzyme in combination with 20 pmol forward primer NS3F (33) and reverse primer NS3R (7995’-CCGAACTGGTACGGTA-3’724) were mixed with nuclease free water to a total volume of 50µl reaction following addition of 10µl denatured (5 minutes at 95 °C) RNA. The reaction mix was subjected to initial incubation of 50°C for 30 minutes, 94°C for 7 minutes followed by 35 PCR cycles: 94°C, 10 seconds; 52°C, 30 min; 68°C for 1 minute supplemented by a final extension for 68°C for 5 minutes. An EEV specific nested re-al-time PCR using forward primer forward (NS3nF) (1985’-GGDGCRGATGARTGTGATAA-3’217) and reverse (NS3nR) (6165’-TTKCTAATYCTATCCGCGTTC-3’596) primer was performed using TaqMan™ assay (Roche) on the Roche LightCycler® 2.0. Each reaction consisted of TaqMan™ Master Mix, 20pmol per primer, 10pmol of the EEV specific FAM-TaqMan® probe (4985’-FAM-TGCGGTYTGATGARATGGAG-3’517) and 2µl of first round product. Nuclease free water was added to the reaction to give a total volume of 50µl. Initial incubation was at 95°C for 10 minutes followed by 45 PCR cycles: 95°C for 10 seconds, 52°C for 1 minute and 72°C for 1 second and cooled at 4°C for 30 seconds.”
Table 1, why are there only 107 positive EEV cases considered in the table, while 111 positive animals were reported L173. What does « SUD » means ? Horses with neurological symptoms are overrepresented in the cohort, could the authors discuss the bias originating from it during data analysis and the confidence about the prevalence of neurological signs in EEV positive animals.
- For the statistical analyses only horses that were documented as having clinical signs were included in the analyses. The horses that had no clinical information (EEV +: n=4; EEV -: n=52) upon submission were excluded from analyses. Therefore, 111 horses tested positive for EEV but 4 were submitted without any clinical information even though we tried to get this information. It is also stated under statistical analyses “Excluded from the OR calculations were animals with no clinical information (n=37), animals that suddenly and unexpectedly died (SUD) (n=9) or aborted fetuses (n=10). The 95% Confidence Intervals (CI) were calculated where relevant.”
- “Sudden and unexpectedly died (SUD)” footnote added to table
- This is part of an ongoing surveillance to determine the etiology of neurological cases in animals, so samples are send biased based on neurological signs and to a lesser extend respiratory, fever etc. “Samples were collected throughout South Africa as part of nationwide passive surveillance of horses to determine the etiological agents responsible for neurological, febrile, or respiratory signs or sudden unexplained death (SUD).” Samples are sent in by veterinarians that list all clinical signs. 51% of cases had neurological signs which were lower than the rest of the cohort suggesting although neurological signs are frequently detected it is less common than in negative cases which are associated with other pathogens such as WNV. The case fatality ratio is only 9% relative to 35% for WNV.
Lines 185-189 : would be interesting here to draw a map and a temporal analysis of EEV cases in South Africa.
- I have included a map of South Africa indicating sample submission and positive EEV cases
Figure 2 is incomplete (clades A and B missing). It would be interesting here to indicate which EEV serotypes have been evidenced in horses and mosquitoes and from which region ; a more detailed analysis would allow to map the diversity of EEV strains in the different regions of South Africa.
- Apologies for this, the clade A/B was just cut of in the pasted version of the figure, in the original PDF uploaded it indicates Clade A/B. I have update and revised the figure
- Unfortunately, we do not have the serotype information for the midges and horses in figure 2 as it is the NS3 gene regions and not VP2 or VP5.
Round 2
Reviewer 2 Report
lines 338-386 - rewrite this part and correct the manuscript and figures as I asked before
The authors did not understand the point I was trying to point out about the names and numbers of the EEVs at all.
The problem with Howell et al 2002 is that they numbered the serotypes based on the alphabet of the names given to the serotypes and it has nothing to do with the date of first isolation. What they now wrote confuses the issue even more.
E,g. Cascara was the name of the horse which became referred to as EEV-1; Kaalplaas is the name of a farm where EEV-3 was first found, as is Bryanston and Kyalami.
This correction is really important for the EEV field.
